# Image-to-Image Translation-Based Structural Damage Data Augmentation for Infrastructure Inspection Using Unmanned Aerial Vehicle

Gi-Hun Gwon [1], Jin-Hwan Lee [1], In-Ho Kim [2,*], Seung-Chan Baek [3] and Hyung-Jo Jung [1,*]

[1] Department of Civil and Environmental Engineering, Korea Advanced Institute of Science and Technology, Daejeon 34141, Republic of Korea; irlgns@kaist.ac.kr (G.-H.G.); archi_tensai@kaist.ac.kr (J.-H.L.)
[2] Department of Civil Engineering, Kunsan National University, Gunsan 54150, Republic of Korea
[3] Department of Architectural Design, Kyungil University, Gyeongsan 38428, Republic of Korea; baeksc@kiu.kr
* Correspondence: inho.kim@kunsan.ac.kr (I.-H.K.); hjung@kaist.ac.kr (H.-J.J.)

**Abstract:** As technology advances, the use of unmanned aerial vehicles (UAVs) and image sensors for structural monitoring and diagnostics is becoming increasingly critical. This approach enables the efficient inspection and assessment of structural conditions. Furthermore, the integration of deep learning techniques has been proven to be highly effective in detecting damage from structural images, as demonstrated in our study. To enable effective learning by deep learning models, a substantial volume of data is crucial, but collecting appropriate instances of structural damage from real-world scenarios poses challenges and demands specialized knowledge, as well as significant time and resources for labeling. In this study, we propose a methodology that utilizes a generative adversarial network (GAN) for image-to-image translation, with the objective of generating synthetic structural damage data to augment the dataset. Initially, a GAN-based image generation model was trained using paired datasets. When provided with a mask image, this model generated an RGB image based on the annotations. The subsequent step generated domain-specific mask images, a critical task that improved the data augmentation process. These mask images were designed based on prior knowledge to suit the specific characteristics and requirements of the structural damage dataset. These generated masks were then used by the GAN model to produce new RGB image data incorporating various types of damage. In the experimental validation conducted across the three datasets to assess the image generation for data augmentation, our results demonstrated that the generated images closely resembled actual images while effectively conveying information about the newly introduced damage. Furthermore, the experimental validation of damage detection with augmented data entailed a comparative analysis between the performance achieved solely with the original dataset and that attained with the incorporation of additional augmented data. The results for damage detection consistently demonstrated that the utilization of augmented data enhanced performance when compared to relying solely on the original images.

**Keywords:** image-to-image translation; generative adversarial network; data augmentation; image generation; damage detection

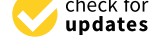



## 1. Introduction

In recent years, the application of UAVs equipped with advanced image sensors for infrastructure inspection has garnered significant attention in the field of maintenance and structural health monitoring [1–3]. This emerging research area aims to exploit the advantages offered by robot and image sensor-based inspections, notably their remote control capabilities and enhanced accessibility, surpassing the limitations of conventional visual inspection methods [4,5]. By enabling more efficient and cost-effective inspections of large-scale structures within reduced time frames, these advanced techniques hold the potential to revolutionize inspection practices. Furthermore, employing robots for

inspections can mitigate human safety risks, as they eliminate the need for personnel to operate in hazardous environments. Through image processing and the analysis of inspection data, these methods promise objective and reliable assessments, avoiding the subjectivity often associated with human inspector expertise. The primary objective of such inspections is to accurately evaluate the condition of individual structural elements and incorporate newly identified changes into past assessment reports, ensuring structural assets' safety and compliance with service requirements. Despite the promising advancements, effectively identifying damage through visual means remains a challenge, necessitating continuous research and development to fully integrate and optimize UAV-based inspection for comprehensive structural assessment [6].

The primary objective of UAV-based inspection is to accurately identify and quantify damage within captured images. This process involves exploring the structural space and capturing images of either the entire area or specific vulnerable regions. Given the abundance of images acquired during inspections, manual damage detection is impractical, necessitating automation. Existing image processing techniques have limitations in handling noisy or rough surface images and detecting various types of damage [7–9]. To overcome these challenges, researchers have transitioned to deep learning algorithms based on convolutional neural networks (CNNs). These algorithms continuously learn from labeled image datasets, enabling automated and efficient damage detection, surpassing traditional methods in terms of time and efficiency. Research has demonstrated the suitability of deep-learning-based algorithms for image-based damage detection and classification, with increasing adoption in civil infrastructure inspection automation. Gao and Mosalam [10] proposed a CNN-based deep learning model for structural damage recognition. Deep-learning-based strategies have been validated for damage status determination, level assessment, and type determination. Modarres et al. [11] proposed a CNN-based deep learning model for identifying the existence and type of structural damage. It was shown to be effective in classifying damage of various sizes or shapes and invariant to image size, location, and noise. The performance was verified against several other machine learning algorithms on real concrete structures. In addition, the deep-learning-based damage detection performance reached a level exceeding that of humans with high reliability and accuracy [12,13]. Among the various processes for the automation and practical application of UAV-based structural monitoring, research on imaging damage detection has become a major focus.

To harness the full potential of deep neural networks (DNNs) and minimize manual interventions, it is imperative to amass extensive and diverse datasets for training models with strong generalization capabilities. Data augmentation acts as a conduit to infuse variety into the distribution of training datasets, efficiently exploiting the available data [14–17]. While common data augmentation methods like random flipping, rotating, and cropping have demonstrated their ability to bolster the resilience of trained DNNs, they possess limitations in encapsulating the authentic diversity seen in real-world structures. This includes the variations in building attributes such as size, shape, and color, consequently curbing these methods' efficacy [18–20]. In response to this issue, researchers have introduced innovative data augmentation strategies that leverage synthetic images—either generated by computers or data-driven [21,22]. These synthetic data augmentations offer a means to introduce a broader spectrum of diverse and lifelike scenarios. This allows DNNs to adapt more effectively to the intricacies of real-world structures, thus enhancing their performance across various scenarios. The inclusion of a broader range of data augmentation techniques has the potential to enhance the resilience and adaptability of DNN models. This, in turn, can lead to improved performance and efficiency in automated inspection procedures for the robotic monitoring of civil structures. However, this data augmentation method requires a physically based structural model in virtual space. In addition, although structures can be created in a more accurate virtual environment through numerical analysis, direct application to real structures has limitations that can only be applied in situations similar to specific structures. Integrating a wider array of data augmentation methods has the potential to enhance the resilience and adaptability of DNN models, thereby leading

to heightened efficiency and efficacy in automating inspection procedures for civil structures through UAV-based monitoring. Nonetheless, this approach to data augmentation demands a physically grounded structural model situated within a virtual domain. Moreover, despite the potential for the more precise creation of structures in a virtual environment via numerical analysis, directly transferring these findings to real-world structures comes with restrictions. Such an application may only be viable in scenarios closely resembling specific structures.

Data-driven augmentation methods based on GANs [23] offer an innovative approach wherein dataset distributions are implicitly learned. This technique employs a GAN model to generate images, addressing data scarcity with fewer resources. Data augmentation methods using GANs have been proposed in various fields, including medicine, construction, and transportation [24–28]. GANs engage in a competitive interplay between a generator and discriminator, resulting in the creation of photorealistic composite images by distinguishing genuine from synthetic images. This strategy widens the range of structural transformations, enabling the model to capture the characteristics of various structures, thereby producing more authentic and diverse scenarios. Our objective is to enhance the efficiency and reliability of robot-driven structural monitoring. This is achieved by cultivating more complex and lifelike image datasets through GAN-driven data augmentation. This approach has the potential to not only mitigate challenges arising from limited data but also enhance the adaptability of automated monitoring systems in the context of civil structures.

In this study, a data augmentation methodology for structural damage images based on image-to-image translation is proposed. Our approach was structured around two main objectives. First, a model was constructed that generates realistic damage images by learning the transformation between annotated and actual images, where annotated images serve as input and actual images are used as output. This allowed for the creation of synthetic images that closely resemble real-world scenarios. Second, the copy-blob method was employed to generate random annotated images enriched with prior knowledge, incorporating elements like piers; decking; and various types of damage such as cracks, peeling, and exposed rebar. These newly generated annotated images were subsequently fed into the established GAN model to produce new training images that simulate real-world scenarios. Three datasets were employed in our study: virtual data, a publicly available collection of cracks, and post-earthquake structural damage images. The performance of our method was systematically evaluated within an existing deep-learning-based damage detection framework, with variations in the frequency of newly generated data utilization. This novel structural damage data propagation method effectively addresses the scarcity of training data for computer-vision-based damage detection, offering the advantage of direct utilization in training without requiring additional image processing or labeling efforts. Furthermore, our performance comparisons consistently demonstrated that incorporating newly generated data yielded improved damage detection outcomes compared to using the original dataset. The remainder of this article is arranged as follows. In Section 2, an overview of the research background related to GAN-based data augmentation techniques is provided. In Section 3, the data augmentation procedure using an image-to-image translation model is detailed. To assess the effectiveness of the proposed approach, experimental tests are presented in Section 4, and finally, the study is concluded in Section 5.

## 2. Related Works

### 2.1. Generation of Structural Damage Images

Numerous studies have explored the creation of synthetic structural damage data to enhance the assessment of structural health. Some have focused on domain adaptation-based image generation, addressing shifts in materials, imaging conditions, and environmental factors. For instance, Weng et al. [29] conducted various domain adaptation tasks to generate images in new environments using crack damage images. They employed DACrack, an unsupervised-learning-based domain adaptation model, to detect anomalies in new domain data. Furthermore, Liu et al. [30] developed a domain-adaptive technique for crack detection

on pavement surfaces, achieving high-performance crack recognition and localization in the target domain through comparisons across diverse road pavement crack datasets.

In addition, GAN-based image generation techniques have been instrumental in automating and enhancing damage inspection and analysis within structural health monitoring. The Damage-T GAN [31] rapidly converts real crack images into numerical damage contours, aiding in swift damage stage determination using reinforced concrete beam datasets. The CrackGAN [32] was proposed to generate ground-truth images of real crack images, employing an asymmetric U-shaped generator network to delineate crack area detection within crack image data. Moreover, various studies have advocated for the use of synthetic images of damaged structures to create large and diverse datasets for more accurate damage identification. CFC-GAN [33], for instance, employs a GAN to predict crack progression on road surfaces, facilitating proactive road maintenance and safety measures by forecasting crack development over time. On the other hand, EIGAN [34] is an unpaired image-to-image translation model that generates realistic composite images of damaged structures based on intact images, leveraging unpaired datasets of damaged building images taken after actual earthquakes and undamaged buildings. The control of damage severity enables the reproduction of various damage scenarios within the images. These research endeavors have significantly contributed to the effective detection and prediction of structural damage. Image generation technology is expected to play a pivotal role in advancing structural health assessment and safety management.

### 2.2. Data Augmentation

Data augmentation is a basic strategy that plays an important role in improving model performance in deep learning. This involves a variety of techniques to artificially extend the training dataset by creating modified versions of the original data. The main goal of data augmentation is to increase the diversity of training examples, allowing deep learning models to generalize better to unseen data. By utilizing augmented data to train the model, it sees a wider variety of examples, making it more robust to variations and reducing over-fitting. Another important aspect of data augmentation is that it helps address class imbalance. In classification tasks, one class may have significantly fewer samples than another class, which can result in a biased model. Data augmentation balances the class distribution by generating additional samples for under-represented classes, ensuring that the model learns each class equally. Data augmentation also improves the robustness of the model. This exposes the model to many different variations of the input data, making it more effective at handling situations where input conditions may vary in the real world. Basic data augmentation techniques, such as geometric transformations (i.e., rotation, scaling, flipping) and transformations (e.g., color, brightness, and contrast changes), are emphasized to improve model performance, making data augmentation one of the most widely used basic learning strategies [35].

Geometric transformations, color shifting, and other data augmentation techniques have proven effective in deep learning model performance through impactful variations. In addition to these techniques, advancements in the field of object detection and segmentation have led to the development of more intricate data augmentation methods. For instance, CutMix [36] involves cutting a portion of one image and merging it with another, thereby blending data from multiple sources. Similarly, the copy-paste method [37] entails copying objects or regions of interest from one image and pasting them onto another. This approach aids deep learning models in recognizing objects and performing object detection in diverse environmental contexts. In the field of civil engineering, Jamshidi et al. [38] explored the enhancement of an automated visual inspection system for concrete structures, focusing on detecting surface defects like cracks. They employed transfer learning, fine-tuning a pre-trained U-Net model with a synthetic dataset generated using CutMix data augmentation, and introduced a temporal data fusion technique for sequential images to improve segmentation network performance, resulting in a significant increase in the F1 score and mIoU by 28.4% and 22.2%, respectively. Çelik et al. [39] present a novel sigmoid-optimized

encoder-decoder network tailored for crack segmentation. They emphasize the substantial performance enhancement achieved through the innovative approach of copy-edit-paste transfer learning.

Furthermore, GAN-generated images can be utilized for data augmentation, particularly when generating new data not present in existing datasets. For instance, in cases where parts of an image are obscured or damaged, GANs can predict and complement those missing portions. Additionally, GANs can generate images from various styles, angles, and environments, enhancing a model's adaptability to diverse scenarios. GAN-based data augmentation proves valuable in enhancing model performance across various fields, especially in situations where data acquisition is challenging. Dunphy et al. [19] validated the use of synthetically generated images from GANs for multi-class damage detection on concrete surfaces. Their research indicated that the average classification performance for hybrid datasets decreased by approximately 10.6% and 7.4% for validation and testing datasets, respectively, when compared to models trained solely on real samples. Li et al. [40] introduced a method for synthesizing high-resolution concrete damage images using a conditional generative adversarial network (CGAN) [41] to address the challenges of manual image collection. The GAN-based synthesized images proved reliable and suitable for training deep-learning-based concrete damage detection networks. However, there are two key challenges. Firstly, the absence of a systematic framework for generating data, and secondly, the limited validation on structural damage images beyond the proposed datasets, such as crack patch images. Hence, there is a need for a systematic framework to directly utilize GAN-based image generation techniques for data augmentation, along with validation on a broader range of data.

## 3. Proposed Methodology

This section provides a comprehensive summary of the entire process for image-to-image translation-based structural damage image data augmentation, as depicted in Figure 1. In the first step, an image-to-image translation model is trained using the original RGB image and its corresponding labeled image. The input image in this step is a labeled mask image, while the target image is an RGB image paired with the semantic segmentation result of the labeled image. The trained image generation model plays a pivotal role in data augmentation, allowing the generation of new RGB images matching the newly provided labels according to the format described in the second stage.

In the second step, a labeled image is reconstructed to augment the damaged data by leveraging prior knowledge of the structural damage. Typically, damage like cracks and spalling is found on structural components such as walls, slabs, and columns, while non-structural elements like windows remain unaffected. Leveraging this information, a new damage mask is assigned to the existing labeled image for pre-processing and generating a structural damage image. Initially, original mask images are input based on the dataset specifications. Then, it is possible to identify areas that may have damage through the labeled RGB values of the original mask image. For example, in Figure 1, pink areas represent shear walls and columns where damage may be present in the building structure, while gray areas for windows and the black background indicate where damage cannot exist. Based on this information, extraction is prioritized for areas where damage may exist. Afterwards, a single damage type such as cracks or spalling present in the damage sample dataset is applied to the damaged area. Labeled data containing new damaged samples are generated to create a mask image from which new structural damage images can be created. Finally, the image produced by the image-to-image translation model from the first stage serves as the input image for generating a new RGB image that includes the damage.

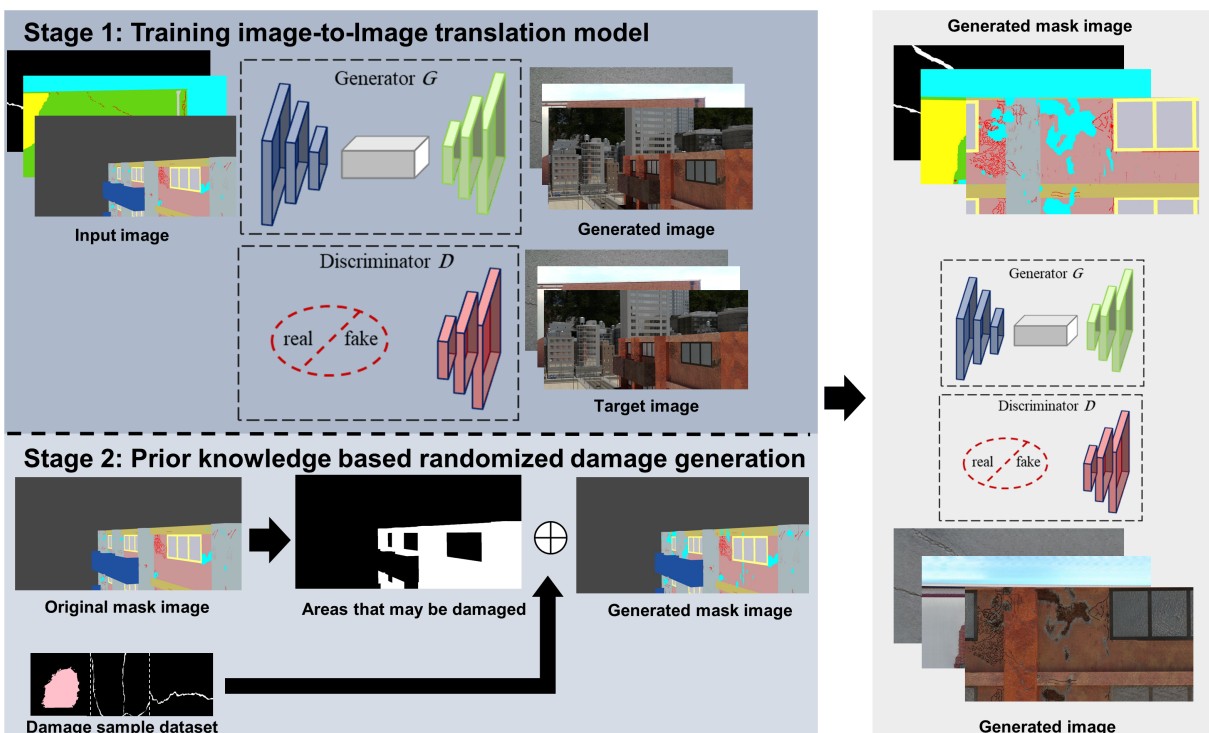

**Figure 1.** Proposed methodology of structural damage data augmentation based on image-to-image translation model.

### 3.1. Stage 1: Image-to-Image Translation Model for Structural Damage Image Augmentation

The image-to-image translation model for structural damage data augmentation was built using the Pix2PixHD [42] architecture and is shown in Figure 2. This model was improved using Pix2Pix [43], which first appeared for image-to-image translation work, as a base model. This model is based on a cGAN and performs image generation by learning the mapping between input images and desired output images. However, it has problems expressing subtle details in high-resolution image generation tasks. The model used in this study produced better results than previous models in the task of generating high-quality, high-resolution images starting from low-resolution images.

The key features of this model are that it uses a coarse-to-fine generator, a multi-scale discriminator, and an improved adversarial loss function to generate high-resolution images. First, the coarse-to-fine generator is composed of two sub-networks. The $G_1$ network is the first step, which considers low-resolution input images and generates low-quality images. In other words, the network responsible for the initial creation stage performs image creation taking into account the overall structure and major characteristics of the image. The results generated here tend to lack detailed information. The local enhancer network $G_2$ generates high-quality images by considering the low-resolution images generated in the previous step. This network improves the quality of the images it produces by learning a mapping to the details obtainable at higher resolutions. Finally, the learning information from both networks is combined to perform the task of generating high-resolution images. This is suitable for obtaining higher-quality results in high-resolution image-to-image translation.

A multi-scale discriminator evaluates an input image at multiple resolution levels and determines whether the generated image is realistic at each resolution level. This contributes to improving the safety and quality of the high-definition image creation process. The goal of the discriminator is to evaluate the difference between the input image and the actual image. Typically, evaluations are made at multiple levels of resolution, such as the original resolution, half resolution, and 1/8 resolution, to take into account different details and structures in each scaled image. All discriminators share the same architecture, but the discriminator that operates at the coarsest scale guides the generator

to produce globally consistent images. In contrast, the finest discriminator focuses on encouraging the generator to map finer information. The multi-scale discriminator can avoid the generation of repetitive patterns in the generated high-resolution images. The multi-scale discriminator computes the GAN loss as follows:

$$L_{GAN}(G, D_k) = \mathbb{E}_{(s,x)}[\log(D_k(s,x))] + \mathbb{E}_s[\log(1 - D_k(s, G(s)))] \tag{1}$$

where $L_{GAN}$ represents the GAN loss; $G$ is the generator; $D_k$ represents the $k$-th scale discriminator; and $s$ and $x$ denote the semantic label map and real image, respectively. Additionally, feature matching loss is calculated by comparing the features between the generated image and the actual image so that the generator can obtain the desired result. The feature matching loss that calculates the feature differences between the generated image and the actual image at each scale is as follows:

$$L_{FM}(G, D_k) = \mathbb{E}_{(s,x)} \sum_{i=1}^{T} \frac{1}{N_i} \left\| D_k^{(i)}(s,x) - D_k^{(i)}(s, G(s)) \right\|_1 \tag{2}$$

where $L_{FM}$ represents the feature matching loss, $D_k^{(i)}$ denotes the feature output from the $i$-th layer of the $k$-th scale discriminator, $T$ is the total number of layers, and $N_i$ is the number of feature elements output from the $i$-th layer. Finally, the comprehensive objective function for the multi-scale discriminator is formulated as follows:

$$\min_G \left( \max_{D_1, D_2, D_3} \sum_{k=1}^{3} L_{GAN}(G, D_k) + \lambda \sum_{k=1}^{3} L_{FM}(G, D_k) \right) \tag{3}$$

where $\lambda$ serves as a hyper-parameter regulating the trade-off between the significance of the GAN loss and the feature matching loss. The multi-scale discriminator is pivotal in ensuring the stable training of the generator for tasks involving high-resolution image generation, ultimately leading to superior-quality outcomes. The learning process of the proposed method is performed as follows. The target generated image of the entire network is an image with structural damage, and the input image is a mask image that matches it. Using multiple generators and discriminators, when a new mask image is input, a high-quality structural damage image suitable for data augmentation is generated.

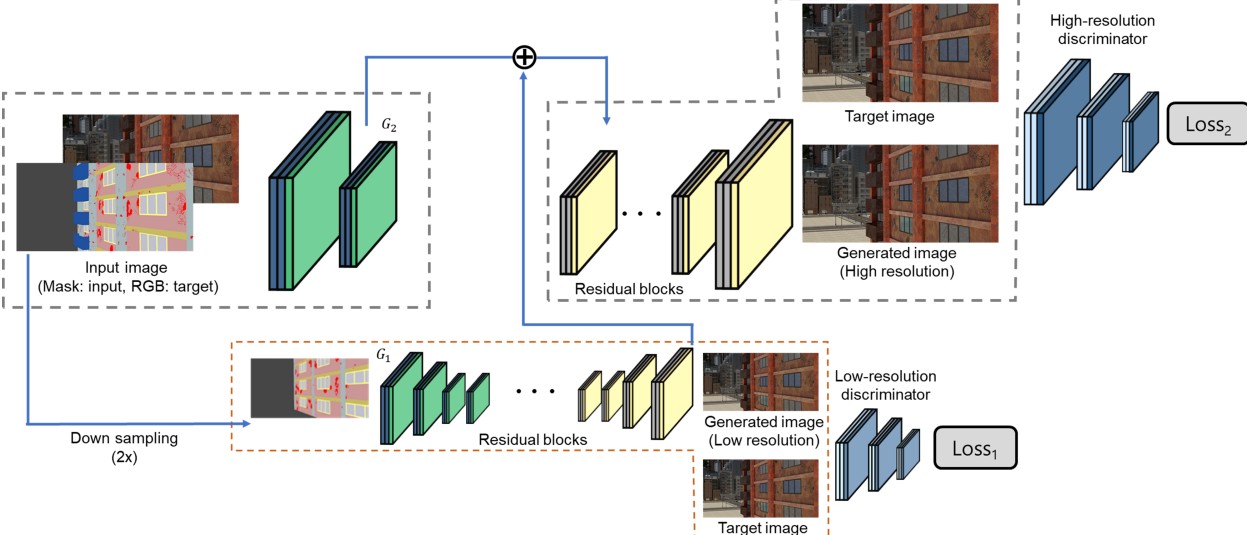

**Figure 2.** Architecture of image-to-image translation model.

### 3.2. Stage 2: Prior-Knowledge-Based Randomized Damage Generation

The primary objective of stage 2 is to generate masks for data augmentation, leveraging expert knowledge to reflect real-world scenarios. For example, in images of concrete cracks acquired at close range, the target crack and the background are typically distinguishable. In this scenario, cracks are likely to exist throughout the background. Therefore, in order to increase the amount of data, it is possible to create an image by adding a mask similar to a crack to the existing mask image. However, when a wider field of view is required, such as images obtained from a UAV for structural damage assessment, more specialized knowledge may be required. The acquired image may include not only the damaged structures of the object, but also the background, extraneous structures, and other elements outside the region of interest. Additionally, the damage present in major structural components such as beams, columns, and walls within the target structure is more important in determining structural risk. Damage to non-structural components such as windows, cladding, and railings can be significant considering the risk of falling debris and maintenance and reinforcement requirements. Therefore, the goal of the proposed method is to generate mask images of the damage present in the appropriate components, taking into account the main damage categories in the dataset.

Figure 3 illustrates the prior-knowledge-based randomized damage generation process from images of spalling and cracks in structural brick cladding components, which were used for the experiments below. Firstly, all RGB images in the dataset are accompanied by corresponding mask images, and the RGB image corresponds to a four-color mask image. The background is represented by blue; non-structural elements are in gray; damage such as spalling and cracks are in yellow and red, respectively; and the intact area of the brick exterior, which is the area where damage can occur, is depicted in green. To create images for data augmentation, areas where damage may occur are extracted from the mask image. In the example data, this refers to the area containing green pixels. Subsequently, the shapes of the damage are extracted from the sample images, such as the cracks and spalling on the right, and are input into the areas where damage can occur, thereby generating new mask images. Through this process, it is possible to generate mask images with the addition of new spalling and cracks from the existing mask images.

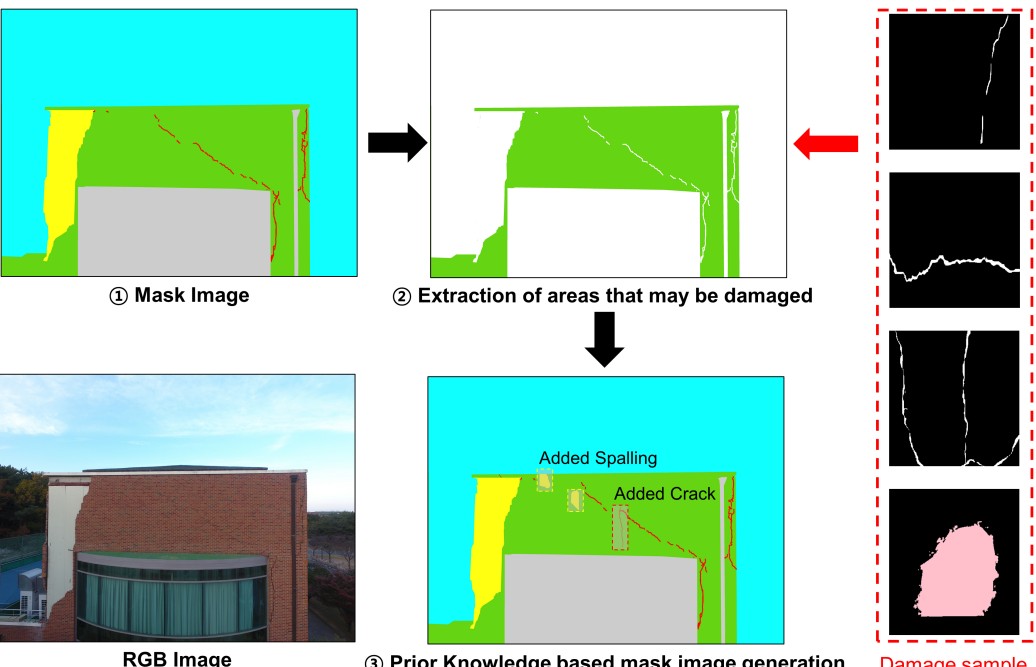

**Figure 3.** Prior-knowledge-based randomized damage generation process.

## 4. Experiments

In this section, the experimental validations of data augmentation using GANs are presented. First, the three datasets used in the experiments are described. These datasets included close-up crack data, earthquake-induced brick cladding structural damage data obtained via UAVs, and post-earthquake structural data constructed in a virtual environment. Afterwards, the structural damage images generated for data augmentation for each dataset are described. The similarity between images generated from prior-knowledge-based mask images using the image-to-image translation model is discussed. Finally, the damage detection results based on the augmented data are discussed.

*4.1. Datasets*

To create images, we utilized datasets covering various scenarios, from basic crack detection to comprehensive building component data, including damage cases. The image size of the entire dataset was set to 1024 in width and 512 in height, considering GPU capacity and high-quality image generation. In addition, from the entire dataset, some damage masks that were not used for learning were used to construct a sample dataset with the original size. Details of each dataset are introduced in the following subsections.

4.1.1. Dataset 1: Public Crack Images

In the field of crack detection for maintenance, numerous researchers have made efforts to construct benchmark datasets specifically designed for crack detection and segmentation tasks, such as DeepCrack and CrackTree [8,44–48]. Kulkarni et al. [49] provided a comprehensive collection of infrastructure crack types covering a wide range of scenarios such as pavement, bridge, and building cracks. Throughout the dataset, crack damage in the mask images is depicted in white (RGB: 0, 0, 0), while the rest of the background is represented in black (RGB: 255, 255, 255), as shown in Figure 4a. For experimental validation in this study, we utilized CFD [44] as the validation dataset, as outlined in the following. We employed 107 images for training purposes and reserved 11 images for testing. Out of the crack datasets, 4972 images, excluding the validation dataset, were employed in stage 2, as mentioned in Section 3.2, for generating new mask images.

4.1.2. Dataset 2: Post-Earthquake Damaged Brick Cladding Structure

The second dataset for the experimental validation of image-to-image translation-based data augmentation pertained to a brick cladding structure affected by the magnitude 5.4 Pohang earthquake that occurred in South Korea in 2017. This dataset comprised a total of 101 images for training and reserved 11 images for testing. The image acquisition for the target structures was performed using DJI's UAV, Inspire 2, equipped with a Zenmuse X5S camera. The damage in this dataset consists of spalling and cracks caused by the detachment of brick cladding, rather than damage to major structural components. Such damage is essential information for assessing the hazard of falling debris and facilitating rapid recovery following an earthquake.

The second dataset, shown in Figure 4b, encompassed more complex scene information compared to the previous dataset. To facilitate data augmentation and damage detection from RGB images acquired by UAVs, mask images representing various elements within the scene were generated. These mask images were designed to distinguish different components, including the background unrelated to the region of interest (RGB: 15, 255, 255); intact exterior material (RGB: 100, 212, 19); non-structural elements (RGB: 204, 204, 204); and damage such as cracks (RGB: 255, 0, 0) and spalling (RGB: 255, 255, 17). Utilizing a multi-class semantic segmentation mask enabled the image-to-image translation model to discern diverse components, which is crucial for accurate texture-based data augmentation.

4.1.3. Dataset 3: Post-Earthquake Structure in a Synthetic Environment

The third validation dataset was the QUAKECITY dataset introduced by Hoskere et al. [21,50]. This dataset consists of images obtained from post-earthquake struc-

tures in a 3D synthetic environment. The primary purpose of this dataset is to provide an ideal platform for developing and evaluating autonomous vision-based inspection algorithms by reproducing various real inspection scenarios and scenes using advanced graphics modeling techniques and physics-based simulations. The original dataset comprised RGB images; images for component segmentation; mask images depicting damage such as cracks, spalling, and exposed rebar; and videos representing the damage state of each component. For the experimental validation of the image-to-image translation-based data augmentation technique, we used RGB images along with component segmentation images combined with a single mask image representing each damage type.

This dataset comprised 432 images for training and 47 images for testing, as shown in Figure 4c. Each image was obtained along a planned path, simulating post-earthquake damaged structures in a virtual environment, similar to what a UAV would capture. The RGB images were accompanied by corresponding mask images, which consisted of background (RGB: 70, 70, 70); structural elements where damage can occur, such as walls (RGB: 202, 150, 150), beams (RGB: 198, 186, 100), columns (RGB: 167, 183, 186), and slabs (RGB: 193, 134, 1); and non-structural elements like window frames (RGB: 255, 255, 133), window panes (RGB: 192, 192, 206), and balconies (RGB: 32, 80, 160). Additionally, damage in the mask images was represented by cracks (RGB: 255, 0, 0); spalling (RGB: 0, 255, 255); and exposed rebar (RGB: 0, 255, 0). The masks in this third dataset were distinguished by ten different pixel colors, representing complex structural images.

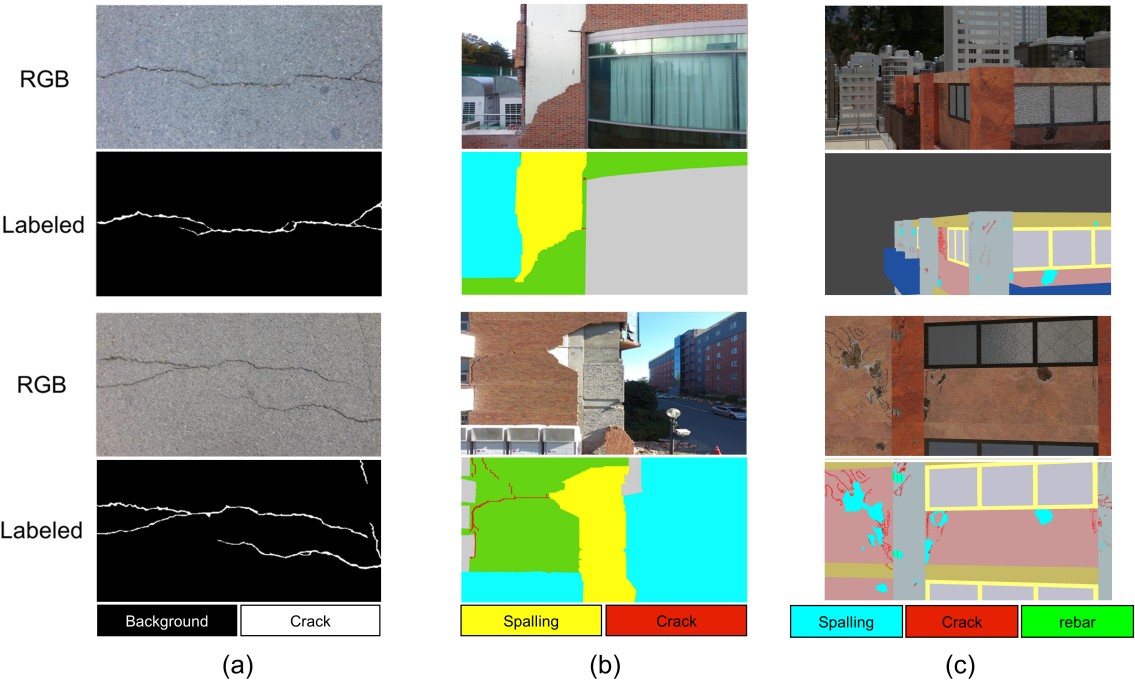

**Figure 4.** Visual examples of each dataset: (**a**) dataset 1, (**b**) dataset 2, and (**c**) dataset 3.

### 4.2. Data Augmentation Results

In this section, we provide a detailed account of the GAN model training process using each dataset and the results obtained by generating corresponding RGB images from the newly created mask images. Training was consistent across all datasets on a single Nvidia RTX 3080 GPU with 10GB of video memory. Each training epoch processed 100 images in about 30 seconds, and a total of 300 epochs was reached. The parameters included a batch size of 1, an Adam optimization momentum weight of 0.5, a feature loss weight ($\lambda$) of 10, and a learning rate of 0.0002. Overall, the training primarily utilized mask images as input data, with RGB images based on pixel information serving as target images. Consequently, as the training progressed, the model enhanced its ability to generate RGB images based on mask information.

### 4.2.1. Data Augmentation Results for Dataset 1

While generating an augmented model for a crack dataset, the performance of the model improves significantly over successive epochs, resembling real footage. Figure 5 shows the image generation results of the model across epochs in the crack dataset. Initially, images generated with masks and a relatively small number of epochs failed to accurately represent the intended damage characteristics. Consequently, these early generated images did not align well with the desired target images. However, as the training process progressed, the output of the model evolved to exhibit similarity to actual images, effectively revealing the damage caused by the mask. With an increasing number of training epochs, particularly after the 200th epoch, the ability to generate images highly similar to the target images improved significantly. The generated images began to display similarities to the actual damage depicted in the target images. Even when compared with the original target image, it was evident that the damage was reasonably recreated. With each epoch, the model refined its understanding of this relationship, gradually enhancing its ability to interpret the masked region and synthesize its damage properties.

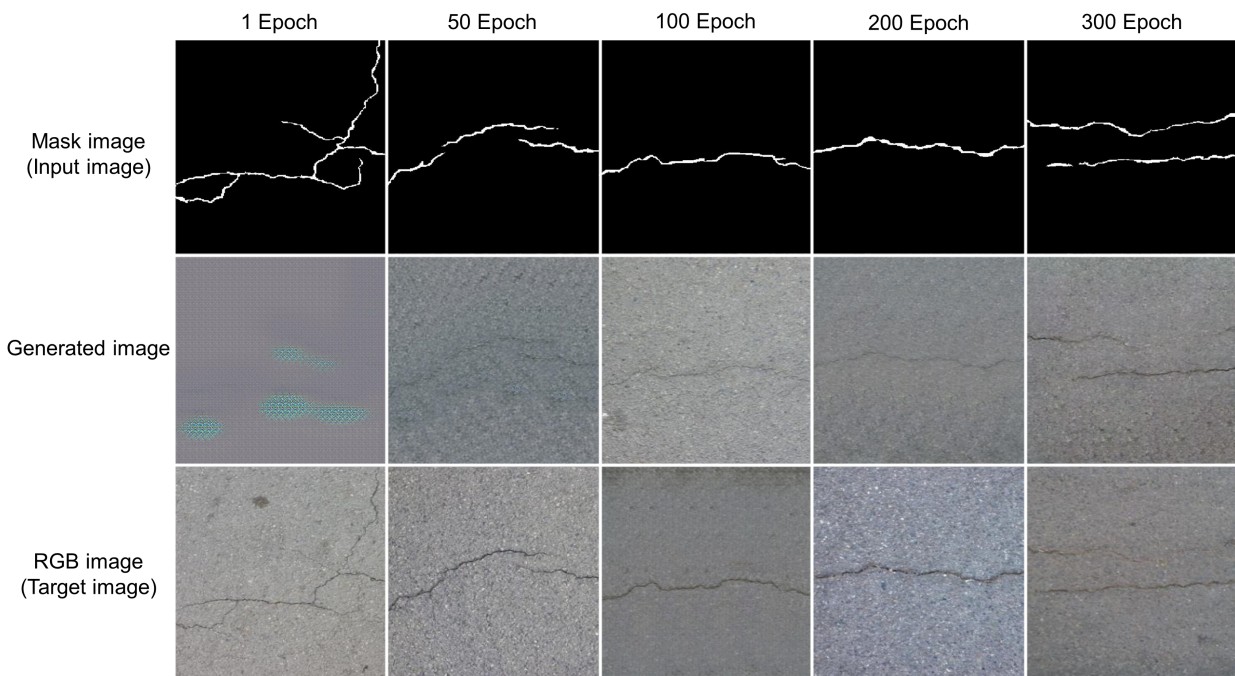

**Figure 5.** Progress in training the RGB image generation GAN model on dataset 1.

In the process of augmenting the crack dataset, a notable opportunity arose with the creation of new masks that encompassed damage instances not previously present in the existing images. These newly generated masks offered a unique avenue for expanding the dataset and injecting a diverse array of damage patterns into the augmentation process. As the augmentation model generated RGB images by pairing these novel masks with existing images, it effectively introduced new damage attributes and scenarios that were absent from the original dataset. This phenomenon is particularly significant because it enables the augmentation process to incorporate a broader spectrum of damage instances that may not have been captured in the initial dataset due to real-world limitations or the limited scope of available images.

In expanding a crack dataset, data augmentation involves generating new images that contain instances of damage that were not previously present in the original image. In addition to the CDF crack data used in the first dataset, the dataset could be expanded by utilizing mask shapes from other benchmark crack datasets. Figure 6 illustrates the augmentation of data generated from the original mask images, resulting in the addition of images from random damage samples. For instance, as shown in Figure 6a, the first added

damage sample of image 1 was a crack mask close to the vertical central axis, which was moved from an actual crack image. The RGB image generated from this mask reflected the degree of damage based on the mask and produced textures reminiscent of those found in actual concrete images. The added mask images, such as that seen in Figure 6b for image 14, demonstrated that reasonable data generation could be achieved by adding complex crack masks to existing simple crack images. Integrating these new damage patterns, not previously present in the original dataset, enhanced the diversity and comprehensiveness of the dataset. This shows that GAN models effectively enhance datasets by learning to synthesize RGB images that represent a wider range of potential real-world scenarios.

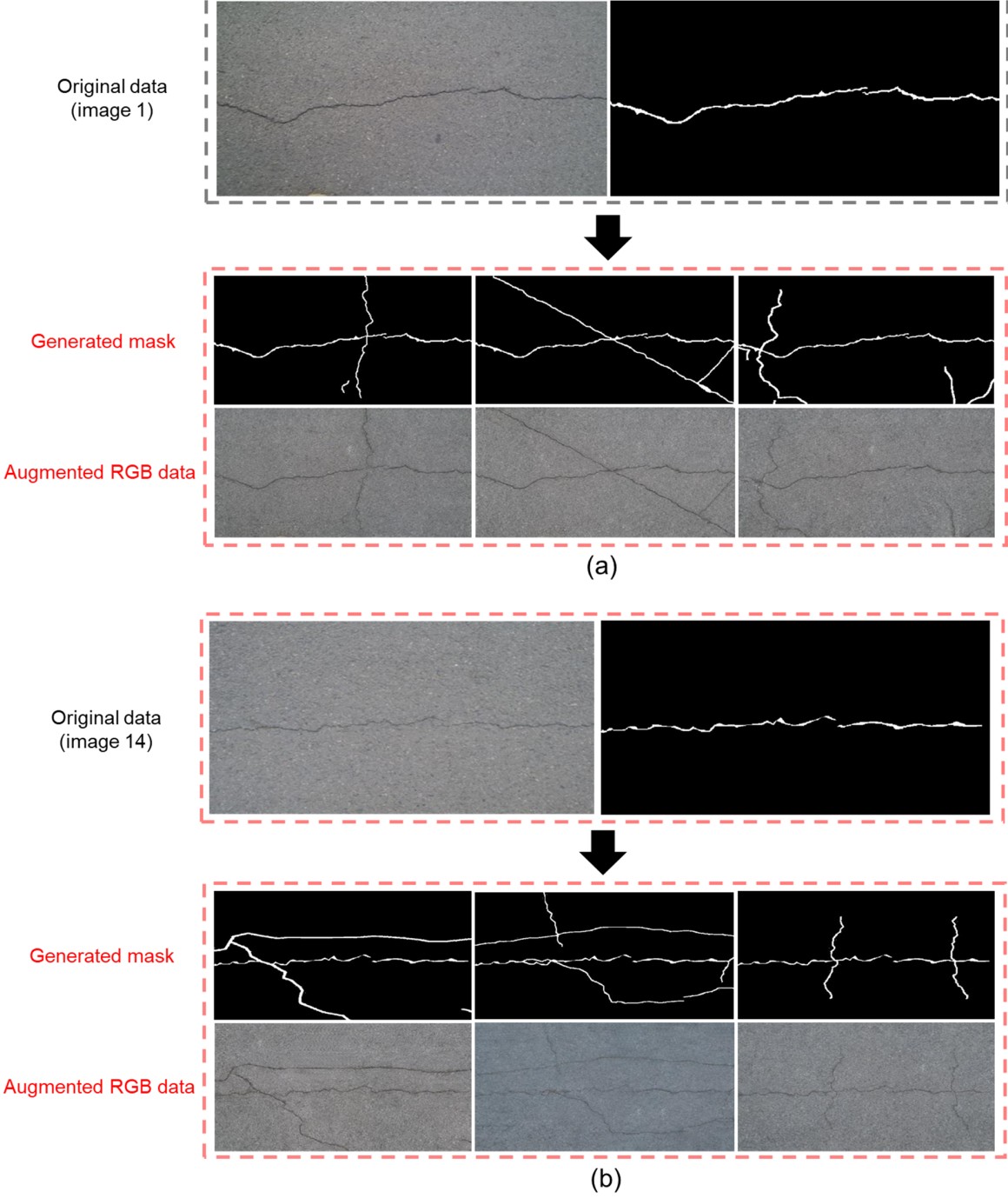

**Figure 6.** Examples of augmented data from specific images in dataset 1: (**a**) image 1, (**b**) image 14.

### 4.2.2. Data Augmentation Results for Dataset 2

The augmentation of the second dataset was performed on images of earthquake-damaged brick cladding structures. This process demonstrated distinct patterns in model performance over time, especially when generating images based on the provided masks. Figure 7 shows the GAN-model-based image generation results according to the learning epoch. Initially, we observed that with a limited number of training epochs, the images generated did not align with the intended damage scenarios defined by the masks. These initial results did not match the desired target images depicting real post-earthquake damage scenarios. However, as the model continued learning over subsequent epochs, notable changes occurred. With an increasing number of training iterations, the augmentation model evolved into a powerful tool capable of generating images highly similar to the actual target images representing the damaged brick cladding structure. After approximately 100 epochs, we observed that some images closely resembled the RGB images of the actual target images. Representation based on specific damage masks, such as spalling and cracks, closely approximated reality. However, the generation of a cyan-colored background was not performed accurately, likely due to the representation of various textures or content under a single label. Nevertheless, since the accuracy of background generation is not crucial for damage detection in target structural images, this issue may not be a significant concern.

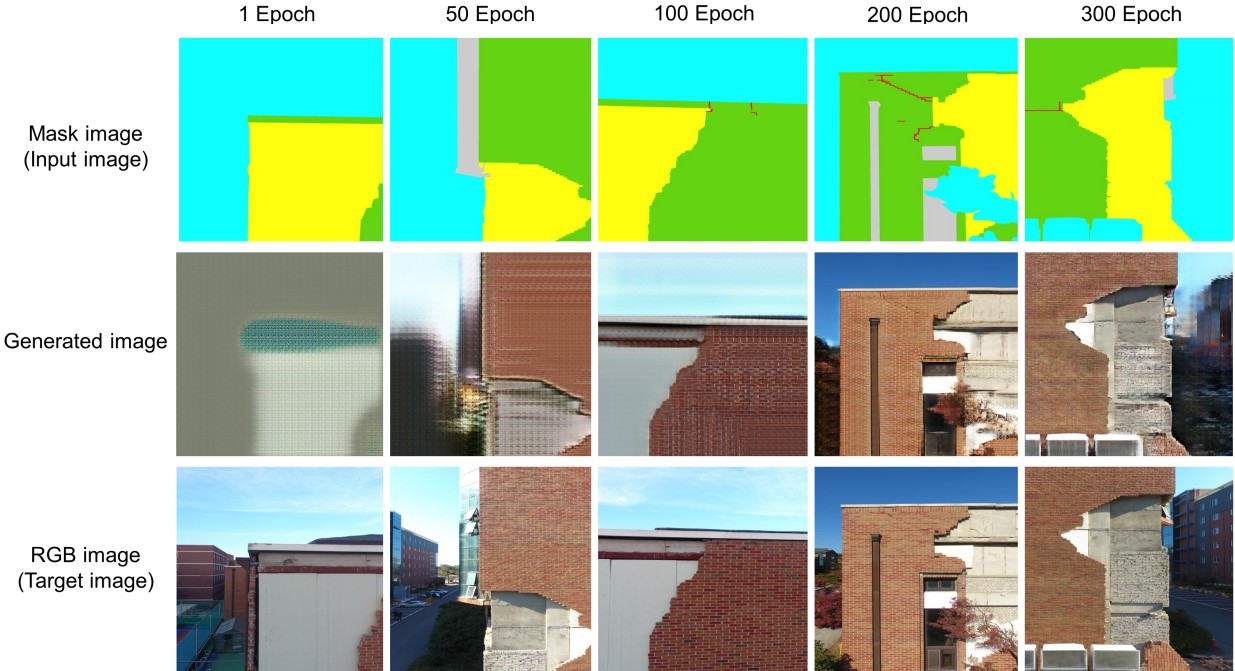

**Figure 7.** Progress in training the RGB image generation GAN model on dataset 2.

In the second dataset, in addition to data samples obtained from a different benchmark crack dataset, we also used crack and spalling damage patterns from the third dataset to generate mask images. In the process of data generation, we extracted areas where additional damage samples could be added and randomly included two instances of spalling and one crack. Figure 8 shows the results of data augmentation using the new mask image from images of damaged structures after the earthquake. As the newly reconstructed masks were input into the image-to-image translation model, they generated RGB images depicting damage attributes that were previously unseen. Figure 8a represents the augmented data from image 3 used in the training of the second dataset, while (b) represents augmented data from image 46. According to the information assigned to each of these target images, the generated images closely resembled the original images but also included information about the added damages. However, there were some issues. Firstly, the background in both the training phase and the generated images was not

clearly represented as in the actual environment. As mentioned earlier, this issue does not significantly affect structural damage detection. Additionally, damage such as spalling, represented in yellow, was clearly expressed, resulting in realistic outcomes. However, there was an issue with the representation of cracks marked in red, as they appeared indistinct and blunt in some images. This was because damage like spalling occupied a significant portion of the overall dataset, while cracks existed in only a few images and constituted a small portion of the total pixels. Due to the label imbalance in the existing deep learning model, similar outcomes could be observed in the image generation models, leading to performance degradation.

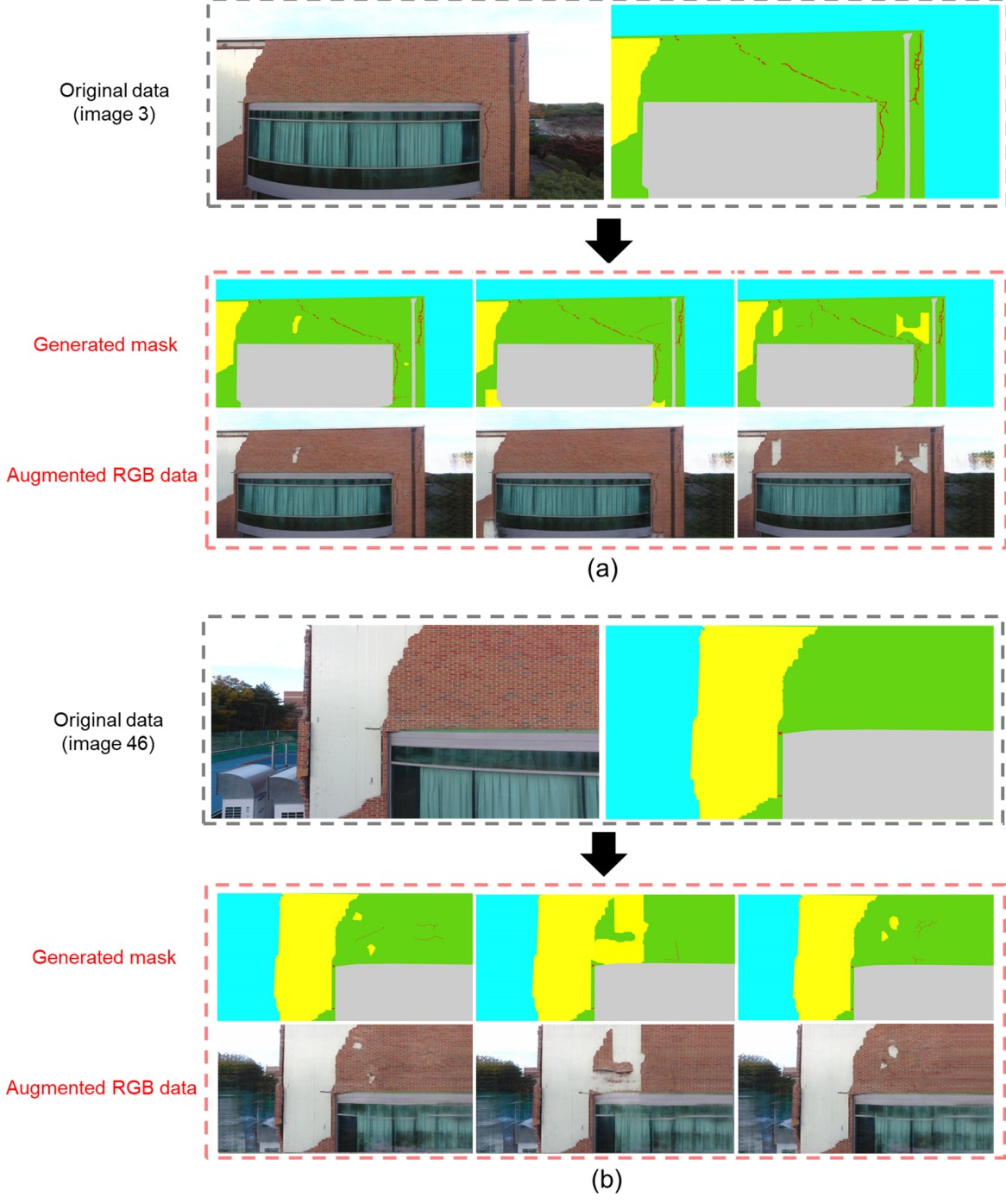

**Figure 8.** Examples of augmented data from specific images in dataset 2: (**a**) image 3, (**b**) image 46.

### 4.2.3. Data Augmentation Results for Dataset 3

In this experiment, data augmentation was performed on virtual-environment earthquake-damaged structural images using an image-to-image translation model for the third dataset. Similarly, to maintain consistency in the training process, the same parameters as those used for the previous datasets were retained. The training of the GAN model for data augmentation, based on the provided masks, revealed a significant improvement in model performance over time. Figure 9 illustrates the results of the image generation process using the GAN model across various training epochs. Initially, due to the limitation of training epochs, the generated images faced situations where they did not closely resemble the intended features defined by the masks. However, as the model continued the training process and the number of training iterations increased, it evolved into a powerful tool capable of closely reflecting actual target images and generating images that depicted damaged structures. This dataset, more complex than the previous ones, encompassed numerous categories, each represented in RGB based on the structural components. Spalling damage were represented in cyan, crack damage in red, and exposed rebar in green. After approximately 100 epochs, we observed that some RGB images closely resembled the actual target images, particularly in the case of images representing specific damage characteristics such as spalling and cracks. However, as with the previous datasets, generating content to match the background was a challenge. Nevertheless, considering that the accuracy of background generation is not a critical factor for damage detection within target structural images, we concluded that the model generated reasonably suitable images for damage detection.

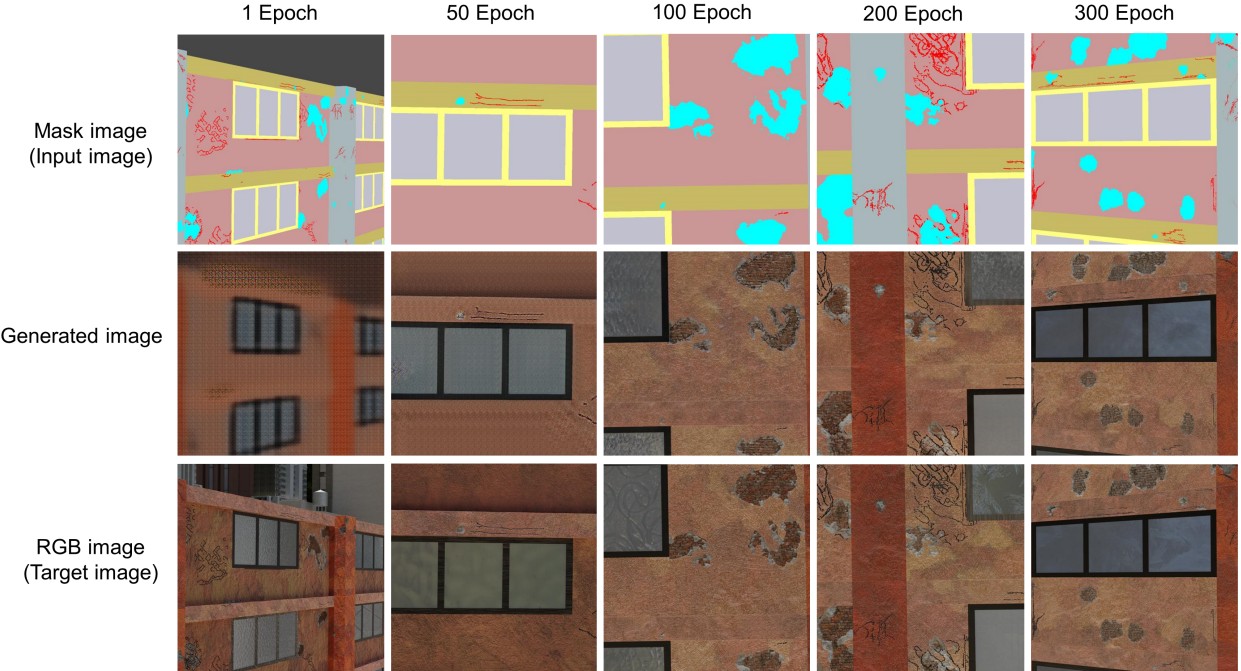

**Figure 9.** Progress in training the RGB image generation GAN model on dataset 3.

In the third dataset, data samples for mask image generation used damage cases from other structural scenarios in addition to the structural scenarios included in QUAKECITY. Similarly to the previous procedure, after extracting the area where there may be damage, one spalling and one crack image were randomly added. Figure 10a showcases RGB images generated from both the original and newly generated mask images for image 1. In contrast, (b) presents a similar comparison for image 4. These newly created mask images incorporated a more extensive range of labels for spalling and cracks when compared to the original mask images. Thus, it can be seen that the results derived from these images were representative in terms of not only the overall components but also various types of damage. This process

led to the creation of images illustrating more severe damage scenarios that were previously unrepresented due to limitations in data collection and the specific range of available images. The integration of these additional damage patterns significantly broadened the scope and diversity of the dataset. When amalgamating these augmented images with the original dataset, the model experienced a substantial improvement in its ability to detect, identify, and comprehensively assess post-earthquake damage scenarios. Consequently, this augmentation process contributed to an enhancement in precision and reliability.

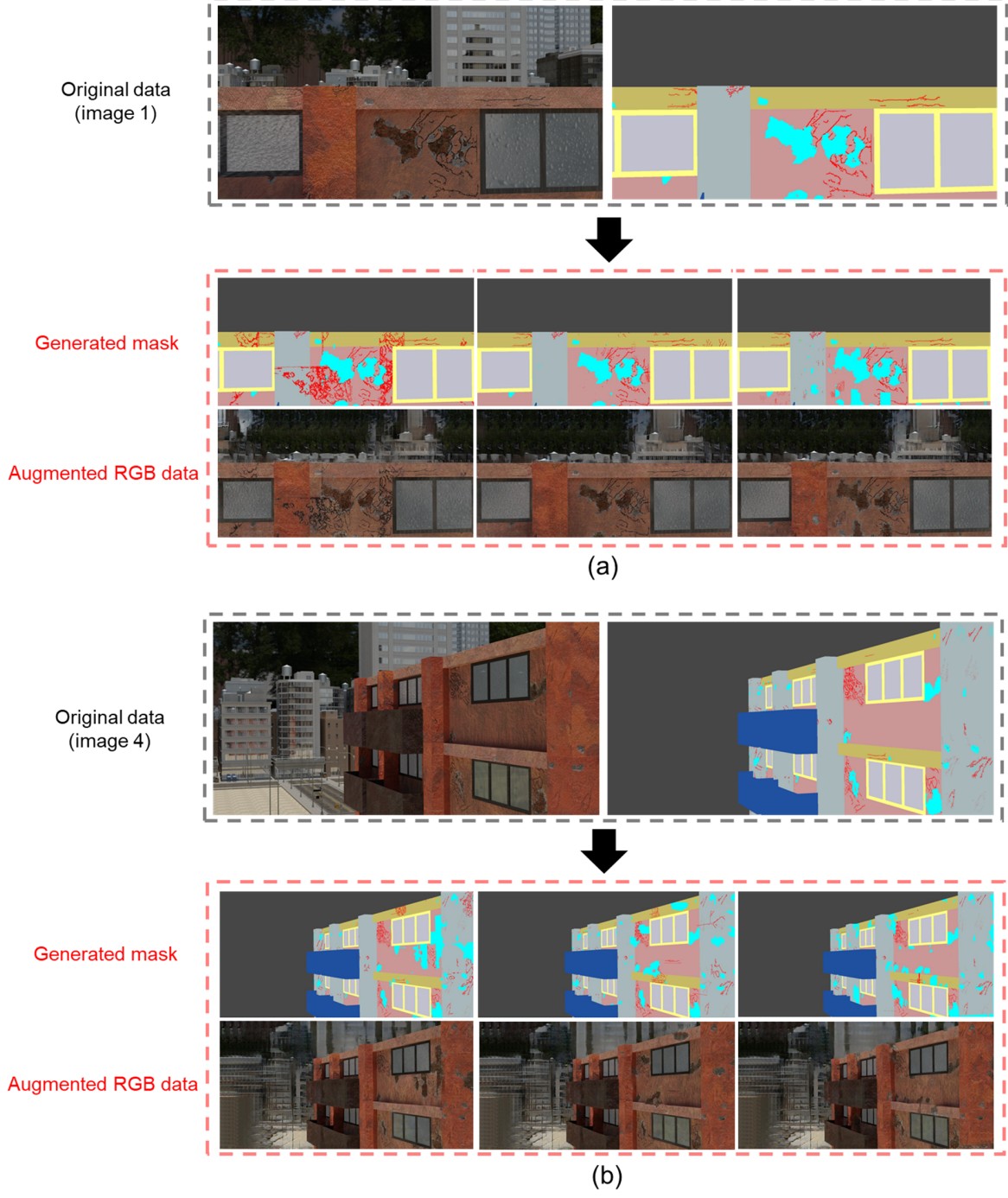

**Figure 10.** Examples of augmented data from specific images in dataset 3: (**a**) image 1, (**b**) image 4.

4.2.4. Quantitative Evaluation

Subsequently, quantitative evaluations were conducted on the images generated by inputting both the training and test data for the three datasets. In this evaluation, the

Frechet Inception Distance (FID) score [51] was the metric used to compare the generated images with real images, quantifying their similarity through a pre-trained Inception v3 model. A low FID score signifies close alignment between the distributions of two images, indicating a high level of similarity.

It is worth noting that FID scores are typically sensitive to image distortions and tend to yield higher scores when the training dataset size is limited. The computed FID scores for the three datasets are presented in Figure 11. Firstly, in Figure 11a, the FID score for dataset 1 was 31.28 at epoch 300. It is noteworthy that the FID score generally decreased with the progression of training, despite some minor fluctuations along the way. In contrast, dataset 2 and 3 exhibited a consistent decreasing trend, as seen in Figure 11b,c. Particularly, dataset 3, which had the highest number of images, boasted the lowest score of 8.22, significantly lower than the other datasets with relatively fewer images.

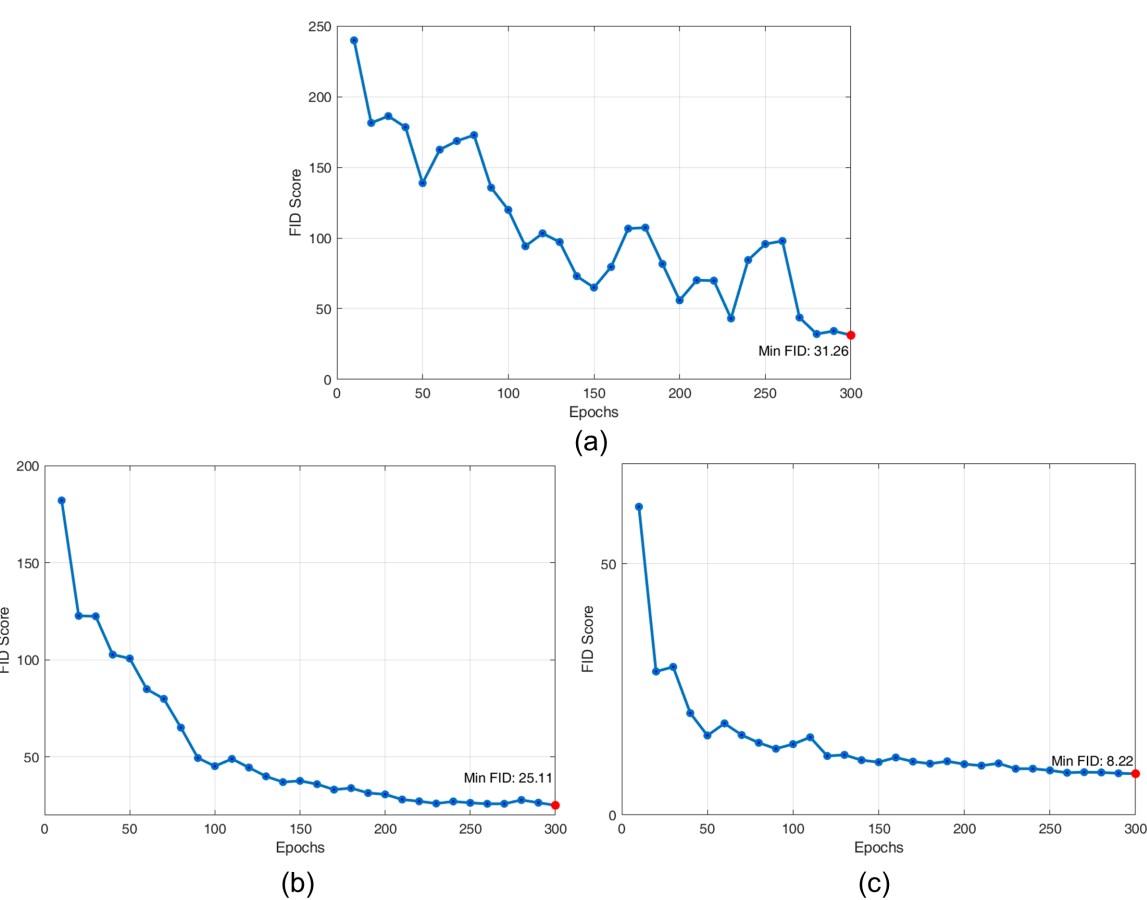

**Figure 11.** FID scores of GAN-generated image datasets: (**a**) dataset 1, (**b**) dataset 2, and (**c**) dataset 3.

### 4.3. Comparison of Damage Detection Performance according to Data Augmentation

In this section, we present a performance comparison of the damage detection model with data augmentation applied. The model utilized for this purpose was based on the image-to-image transformation model used for data generation. Unlike the methods used for data augmentation, this model was designed for structural damage detection by setting structural damage RGB images as input images and mask images as target images. Training for damage detection was carried out for each dataset with 200 epochs, utilizing the same hyper-parameters as described in Section 4.2. For the three datasets, models were built using both the original images and augmented data. The complete data augmentation process involved generating five new mask images from a single mask image, which were then used as input images. In the case of dataset 1, a total of 107 images were used as the base training data, with 11 images designated as validation test data. The augmented data for this dataset amounted to 535 images. For datasets 2 and 3, 101 and 432 images

were used as training data, respectively, with 11 and 47 images designated as test data. The augmented data for datasets 2 and 3 amounted to 505 and 2160 images, respectively. The performance validation of data augmentation was conducted by distinguishing cases where the generated data were used at 150%, 350%, and 500% of the base training data. Figure 12 shows a rough overview of the performance experiment for the damage detection model based on data augmentation.

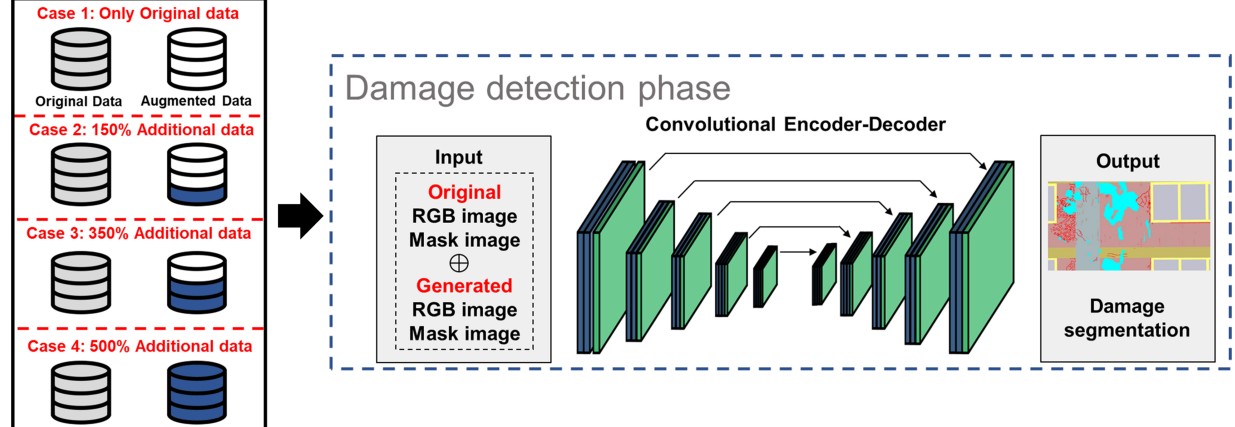

**Figure 12.** Experiment on the performance of the damage detection model with respect to data augmentation.

Table 1 presents the intersection over union (IoU) scores for damage detection based on four different cases of augmented data utilization. Additionally, it includes the improvement in performance compared to using only the original data, considering the ratio of augmented data used. In the case of dataset 1, the model exhibited a slight performance enhancement as more data were retained for crack detection. Interestingly, the highest accuracy in damage detection was achieved when 350% additional data were employed. This indicates that the proposed augmentation techniques generally brought about performance improvements, but increasing the amount of generated data indefinitely did not necessarily translate to an unconditional performance boost. In other words, there seemed to be a convergence point where further augmentation of data could result in diminishing returns.

For dataset 2, which included both crack and spalling classes, slightly different results were observed. Firstly, this dataset predominantly consisted of spalling damage, with relatively fewer images containing information about cracks. Consequently, it showed very low crack detection performance when only the original data were used. The implementation of data augmentation techniques in the crack class did not yield significantly high detection rates but did demonstrate an approximately 200% improvement over the given data. On the other hand, the results for the spalling class showed high detection performance, even with the original data alone, achieving improvements of more than 95%. Here, the utilization of augmented data improved the performance but not dramatically.

The results for dataset 3, which included crack, spalling, and exposed rebar classes, could be explained as follows. Firstly, in the case of cracks, the more augmented data used, the higher the performance improvement, although it eventually converged to a certain detection accuracy. The spalling class exhibited results similar to dataset 2, with no significant improvement in performance with the original dataset, but the performance did steadily increase. The exposed rebar class showed very low damage detection performance with the original data alone, reaching as low as 1.25%. However, with more augmented data, especially when accompanied by a 500% increase in data, it achieved a detection rate of over 30%. These results indicate that the proposed data augmentation techniques contributed to improving the performance of the damage detection model in almost all cases. Regarding common data imbalances in deep learning models, our approach excelled in enhancing performance for classes with limited data while showing modest improvements in sufficient data.

**Table 1.** IoU results for damage detection across different levels of data augmentation in each dataset.

| | | Original Data | 150% Additional Data | 350% Additional Data | 500% Additional Data |
|---|---|---|---|---|---|
| Dataset 1 | Crack | 41.04% | 48.48% (18.13%) * | 49.21% (19.91%) | 49.12% (19.69%) |
| Dataset 2 | Crack | 4.85% | 14.17% (192.16%) | 14.74% (203.92%) | 15.84% (226.60%) |
| | Spalling | 95% | 95.52% (0.55%) | 96.87% (1.97%) | 96.79% (1.88%) |
| Dataset 3 | Crack | 26.99% | 37.20% (37.83%) | 39.02% (44.57%) | 39.61% (46.76%) |
| | Spalling | 79.74% | 80.40% (0.83%) | 81.39% (2.07%) | 81.62% (2.36%) |
| | Exposed Rebar | 1.25% | 7.05% (464%) | 17.29% (1283.2%) | 30.7% (2356%) |

* The values in parentheses represent the IoU improvement rate of the augmentation compared to the case using only the original data.

Following the quantitative evaluation conducted earlier, we now proceed with a qualitative assessment of the damage detection results. Figure 13 illustrates the damage detection outcomes for specific classes within each dataset. Figure 13a showcases the results for dataset 1 based on two RGB images. In the first image patch, it is evident that the cracks on the right side were not correctly detected in all cases when compared to the ground truth (GT). However, when using augmented data, a closer match to the actual damage was achieved compared to using the original data alone. Similarly, for the second image patch, we observed that the augmentation of data gradually improved the detection of cracks on the left side that were initially missed.

In Figure 13b, we present the results of damage detection for dataset 2. Similarly to the quantitative comparison for cracks, it appeared that inappropriate detection was prevalent across all images. However, in all damage detection cases, horizontal cracks were detected to some extent, and with the increase in augmented data, even vertical cracks present in the GT were progressively detected. For the detection of the spalling class, it was noted that in all cases, the model successfully identified damage areas similar to the GT, although when using fewer data, there was a tendency to over-detect regions outside the ground truth. This issue was effectively mitigated with increased data usage. The results for damage in the three classes of dataset 3 are presented in Figure 13c. Regarding cracks, when compared to the GT, the results using only original data seemed relatively discontinuous and failed to estimate the width accurately. Conversely, with the augmentation of data, crack detection aligned more closely with the GT images. In the second column of images in the spalling class, as the amount of augmented data increased, spalling damage was more clearly detected, resembling the GT image. The image of the exposed rebar class provides a clearer comparison. Compared to the GT, the detection model, which used fewer data, failed to detect any damage. On the other hand, using the augmented data, damage was detected in areas somewhat close to those in the GT in the third and fourth cases. To summarize, the qualitative evaluation of the damage detection results across all three datasets underscored the enhancement in detecting various damage classes with continued data augmentation. These improvements were particularly pronounced in scenarios where initial accurate damage detection was hindered due to limited data availability.

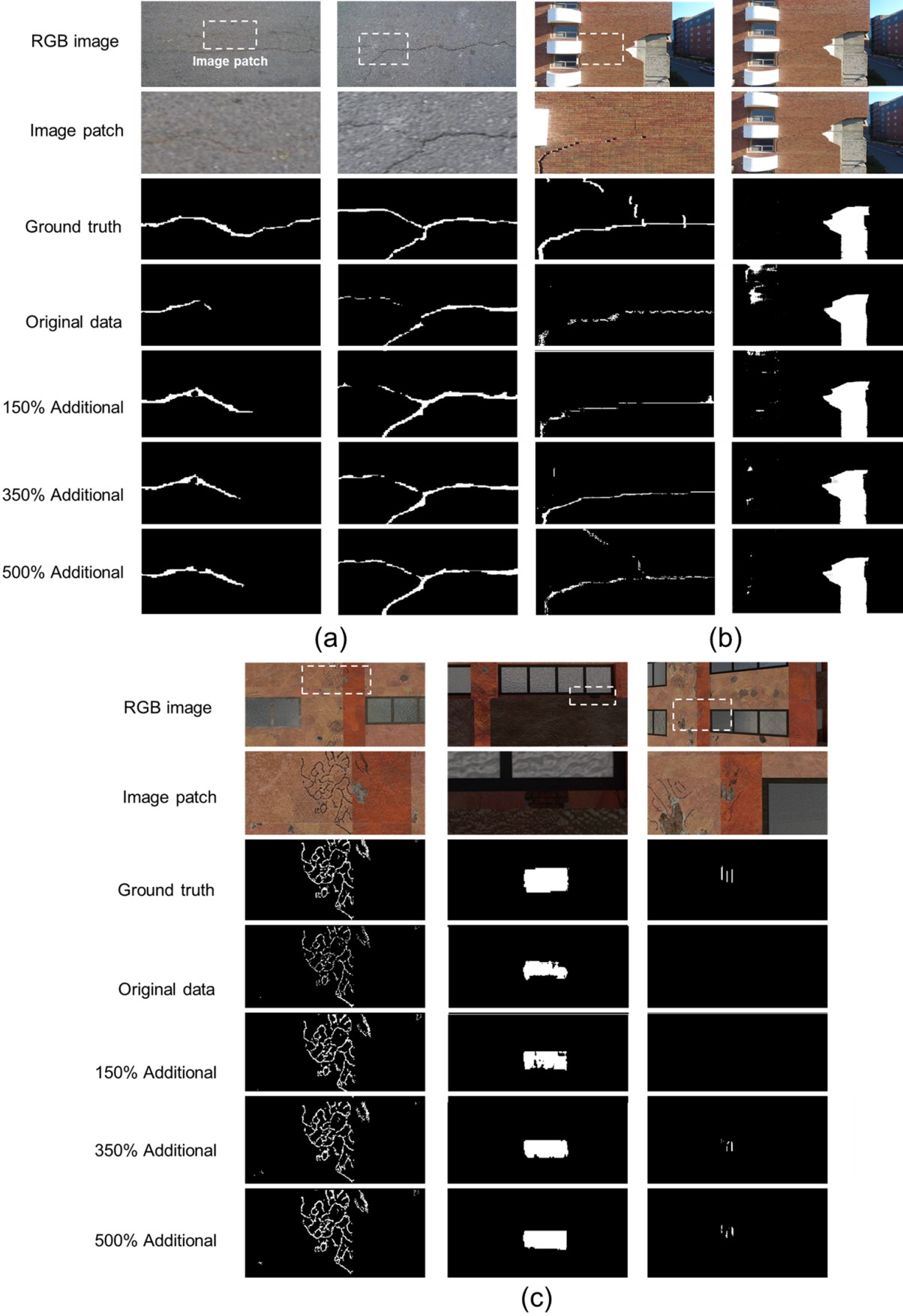

**Figure 13.** Examples of damage detection results according to additional data augmentation: (**a**) dataset 1, (**b**) dataset 2, and (**c**) dataset 3.

## 5. Conclusions

This study effectively demonstrated the potential of data augmentation using GAN models for image-to-image transformation in the field of structural damage assessment and provided an effective solution to problems related to limited data and complex labeling. A significant contribution of our study lies in the development of a novel strategy for generating synthetic structural damage data using a GAN. First, we proposed a domain-specific mask image generation method based on prior knowledge and established a strategy to augment the structural damage mask for augmentation by utilizing domain expertise. Furthermore, the dataset usable for deep learning model training was substantially expanded by training GAN-based image generation models on paired datasets and generating domain-specific mask images for data augmentation. The experimental validation of the data augmentation approach affirmed its capacity to produce integrated images that closely resembled real images yet incorporated novel damage across three datasets, spanning from straightforward crack images to intricate structural representations. Moreover, the experimental validation of damage detection utilizing the newly added data unequivocally illustrated the advantages conferred by the augmented data. When comparing the performance of deep learning models trained with the original data to that of models trained with augmented data, consistent improvements were observed across a variety of datasets, underscoring the capability of our approach to boost the precision and reliability of structural damage detection systems. Additionally, the unique advantage of our proposed method lies in its simplicity and the synergy between mask images and training images used in deep learning. Our data augmentation method augments existing training images with additional damage information based on mask images, eliminating the need for laborious labeling efforts on newly generated images. In conclusion, our approach not only addresses challenges related to data scarcity and complex labeling but also presents an innovative and efficient method for acquiring rich and informative data to train deep learning models. As technology continues to advance, the collaboration between robotics engineering, advanced image sensors, and GAN-based data augmentation holds great potential to advance the field of structural condition assessment and monitoring, ultimately contributing to the safety and longevity of critical infrastructure systems.

**Author Contributions:** Conceptualization, G.-H.G. and H.-J.J.; methodology, G.-H.G. and I.-H.K.; validation, G.-H.G., J.-H.L. and H.-J.J. resources, I.-H.K. and S.-C.B.; data curation, G.-H.G. and S.-C.B.; writing—original draft preparation, G.-H.G.; writing—review and editing, I.-H.K., S.-C.B. and H.-J,J.; visualization, J.-H.L.; supervision, I.-H.K. and H.-J.J.; funding acquisition, H.-J.J. All authors have read and agreed to the published version of the manuscript.

**Funding:** This work was supported by a Korea Agency for Infrastructure Technology Advancement (KAIA) grant funded by the Ministry of Land, Infrastructure and Transport (grant RS-2020-KA156208) and a National Research Foundation of Korea (NRF) grant funded by the Ministry of Science and ICT (NRF 2017R1A5A1014883) and the Research Center for Smart Submerged Floating Tunnel Systems.

**Data Availability Statement:** Not applicable.

**Conflicts of Interest:** The authors declare no conflict of interest.

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
