# Peer review of "Image-to-Image Translation-Based Structural Damage Data Augmentation for Infrastructure Inspection Using Unmanned Aerial Vehicle"

_drones, doi:10.3390/drones7110666_

Round 1
Reviewer 1 Report
Comments and Suggestions for Authors
This paper shows the methodology of infrastructure inspection using Unmanned Aerial Vehicle by proposing a method of Image-to-Image Translation. Authors can improve the Image quality assessment section.
Comments on the Quality of English LanguageAuthors should avoid long sentences.
Reviewer 2 Report
Comments and Suggestions for Authors
This study proposes using Generative Adversarial Networks (GANs) to generate synthetic structural damage data. GANs are trained with paired datasets, and domain-specific mask images are generated to improve data augmentation. Experimental results show that generated images closely resemble real ones and enhance damage detection performance compared to using only original images. In my opinion the paper can be accepted in present form.
- The main question addressed by the research is how to enhance the effectiveness of structural damage detection through the use of Generative Adversarial Networks (GANs) for data augmentation.
- The topic is both relevant and original in the field. It addresses a specific gap by proposing a methodology that leverages GANs to generate synthetic structural damage data for improving the performance of deep learning models in damage detection.
- The research adds to the subject area by introducing a novel approach that combines GANs with structural damage detection, enabling the generation of synthetic data to supplement real-world datasets. This approach may enhance the generalizability and performance of damage detection models.
- Regarding methodology, even if in my opinion it is clearly illustrated, the authors could improve the research following this advised improvements:
- Clear documentation of the training process for the GAN model, including hyperparameter settings.
- A more detailed explanation of the criteria used for selecting domain-specific mask images.
- Addressing potential biases or limitations in the GAN-generated data.
- The conclusions appear consistent with the evidence and arguments presented in the study. The results demonstrate that the utilization of augmented data enhances the performance of damage detection when compared to relying solely on original images.
- The references appear to be appropriate for the context and subject matter of the research.
- Tables and figures are clear and relevant to the research. They effectively support the study's findings and conclusions.
Reviewer 3 Report
Comments and Suggestions for Authors
The manuscript entitled "Image-to-Image Translation based Structural Damage Data Augmentation for Infrastructure Inspection using Unmanned Aerial Vehicle" has made the effort to generate artificial pictures of damaged structures and components using GAN to enrich the database for further training and structural health monitoring purposes. The paper is well-written and the methodology has been described clearly. The novelty of the paper can be questioned moderately as there are lots of other papers targeting the same problem with relatively similar approaches, some of which have been cited in the manuscript. However, the paper has some unique features that make it interesting to potential readers.
the paper can be accepted for publication if the following questions are answered correctly:
1. When training a neural network with images of cracks and damaged structures, how can you enclose infinite crack shapes and patterns? Every damage can have different characteristics and when you put it alongside every independently-designed structure, many damages can be classified as undamaged and vice-versa. Therefore, I believe that the methodology is effective for damages for specific structures and certain well-defined damages such as shear cracks (45 degrees), corner cracks, corrosion, etc. This method and every training-based method fall short of distinguishing every general damage.
2- Generating Data as described in this manuscript can be highly biased to the training-set data. In other words, the method cannot see damages further than the patterns that the real images provided because the artificial images also follow the ground-truth images. How can you go over this bias in your methodology?
Answering these questions in the paper will better show the strengths and weaknesses of the proposed method and will make the paper acceptable for publication.
